# A High Hepatic Uptake of Conjugated Bile Acids Promotes Colorectal Cancer—Associated Liver Metastasis

**DOI:** 10.3390/cells11233810

**Published:** 2022-11-28

**Authors:** Zongmei Zheng, Jiao Wei, Xinxin Hou, Fengjing Jia, Zhaozhou Zhang, Haidong Guo, Fuwen Yuan, Feng He, Zunji Ke, Yan Wang, Ling Zhao

**Affiliations:** 1Academy of Integrative Medicine, Shanghai University of Traditional Chinese Medicine, Shanghai 201203, China; 2Department of Medical Oncology, Shuguang Hospital, Shanghai University of Traditional Chinese Medicine, Shanghai 201203, China

**Keywords:** colorectal cancer—associated liver metastasis, bile acids, Na^+^–taurocholate cotransporting polypeptide, high-fat diets

## Abstract

The liver is the most common site for colorectal cancer (CRC)–associated metastasis. There remain unsatisfactory medications in liver metastasis given the incomplete understanding of pathogenic mechanisms. Herein, with an orthotopic implantation model fed either regular or high-fat diets (HFD), more liver metastases were associated with an expansion of conjugated bile acids (BAs), particularly taurocholic acid (TCA) in the liver, and an increased gene expression of Na^+^–taurocholate cotransporting polypeptide (NTCP). Such hepatic BA change was more apparently shown in the HFD group. In the same model, TCA was proven to promote liver metastases and induce a tumor-favorable microenvironment in the liver, characterizing a high level of fibroblast activation and increased proportions of myeloid-derived immune cells. Hepatic stellate cells, a liver-residing source of fibroblasts, were dose-dependently activated by TCA, and their conditioned medium significantly enhanced the migration capability of CRC cells. Blocking hepatic BA uptake with NTCP neutralized antibody can effectively repress TCA–triggered liver metastases, with an evident suppression of tumor microenvironment niche formation. This study points to a new BA–driven mechanism of CRC–associated liver metastases, suggesting that a reduction of TCA overexposure by limiting liver uptake is a potential therapeutic option for CRC—associated liver metastasis.

## 1. Introduction

Colorectal cancer (CRC) is the third most common malignancy and the second most deadly cancer worldwide [1], ranking as the fourth most prevalent cancer in China [2]. Metastases are the primary cause of CRC–related death. The liver is the most common site of distant spread. Nearly 50% of patients with CRC will develop liver metastases [3]. Hepatectomy, systemic chemotherapy, anti-angiogenic agents, and immunotherapy have substantially improved the clinical prognosis for patients with CRC liver metastasis (CRLM) [4]. However, it still leads to a dismal five-year survival rate and a high recurrence rate [5,6]. Developing strategies for the prevention and treatment of CRLM confronts a considerable challenge.

Developing CRLM is thought to be associated with diverse factors of age, sex, genetics, medical history, location of primary tumors, and lifestyle [7]. Researchers have uncovered that tumor cell-secreted materials (microRNAs, lncRNAs and exosomes, etc.) and specific molecules derived from stromal cells (cytokines, chemokines, and extracellular matrix, etc.) jointly initiate and drive tumor metastasis [8,9]. Dysregulation of liver-dependent endogenous metabolism also contributes to CRLM. A recent study revealed that the aberration of sterol regulatory element-binding protein 2-mediated cholesterol biosynthesis benefits the colonization and growth of metastatic CRC cells [10]. This hints at the importance of liver-specific metabolic change in understanding the mechanisms of CRLM.

Given that high-fat diets (HFD) are strongly associated with CRLM and a worse prognosis with shortened overall survival in patients and animals [11,12], we used HFD feeding to investigate the CRLM–related metabolic signals. Herein, based on metastatic models fed a regular chow diet (CD) or HFD, we found that conjugated bile acids (BAs), e.g., taurocholic acid (TCA), overexposed in the liver, were strongly associated with liver metastases. By treating TCA alone or with an antibody that neutralized Na^+^–taurocholate cotransporting polypeptide (NTCP), TCA overexposure was proven to result in CRLM. Further works revealed that the conditioned medium of TCA–treated liver-residing stromal cells, hepatic stellate cells (HSCs) for example, can promote the metastasis of murine and human CRC cells. This study uncovers a BA–mediated liver-metastatic mechanism, highlighting that controlling hepatic BA uptake is a potential anti–CRLM strategy.

## 2. Material and Methods

### 2.1. Animals and Ethics Statement

Six-week-old male C57BL/6 mice were purchased from Shanghai Jihui Laboratory Animal Care Co., Ltd. (Shanghai, China) and maintained with 10% or 60% of fats in diets (*n* = 18/group) for 4 weeks. All mice were maintained on a 12-h light/dark cycle, 25 ± 1 °C room temperature, 40–60% relative humidity, and free access to water and food. All animal experimental procedures conformed to National Guidelines for Animal Usage in Research (China) and were approved by Shanghai University of Traditional Chinese Medicine (Ethics NO. PZSHUTCM201030020). 

### 2.2. Establishment of CRC–Associated Liver or Lung Metastatic Mouse Models 

For orthotopic transplantation-induced liver metastatic model, 1 × 10^7^ MC38 cells were resuspended in 1 mL sterile PBS, and 100 uL cell suspension was injected subcutaneously into the axilla region of donors. Two weeks later, tumor grafts were surgically removed, cut into 1 mm^3^ pieces and stored in ice cold saline before the surgery. The orthotopic implantation of tumor pieces to cecum was performed referring to the previous method [13]. Briefly, recipient mice were anesthetized with isoflurane and the cecum was clamped out by laparotomy on the left abdomen, then the serosa was gently scraped with the tip of 1 mL syringe and the tumor piece was fixed on the injury site with medical anastomotic glue (Baiyun, Guangzhou, China), last returned the cecum to the peritoneal cavity and closed the peritoneum and skin with 6–0 Vicry1 transmural suture. After 4 weeks, serum, livers, ileum, and tumors were harvested from recipient mice. The pulmonary metastasis model was established as follows. 1 × 10^7^ MC38 cells were resuspended in 1 mL sterile PBS and 100 μL cell suspension was subcutaneously inoculated into the axilla region of donors. Four weeks later, CRC tumor was surgically removed from mice when it reached a maximal volume of 1.5 cm^3^. Mice continued to house for 8 weeks before being sacrificed.

### 2.3. Intervention of TCA or NTCP Neutralizing Antibody

In either liver or lung metastatic models (*n* = 10/groups), TCA were intraperitoneally (IP) injected at the dose of 30 mg/kg for 4 weeks. Referring to a previous study [14], the NTCP neutralizing antibody (USBiological, 350832) was IP administrated to TCA–treated mice twice a week for two consecutives, with PBS as control. 

### 2.4. Detection of Serum TC, TG, LDL and HDL

Serum total cholesterol (TC), low-density lipoprotein cholesterol (LDL–C), and high-density lipoprotein cholesterol (HDL–C) were detected with assay kits purchased from Nanjing Jiancheng Bioengineering Institute and the determinations were conducted according to the manufacturer’ protocol. Briefly, 2.5 μL serum was added to the reagent and incubated at 37 °C for 5 min. Absorbance was measured at 546 nm.

### 2.5. Quantification of Serum Metabolomic Profile 

Metabolomics analysis on serum samples were performed by using the Q300 Metabolite Array Kit (Metabo–Profile) as previously described [15]. The sample was conducted with an ACQUITY UPLC BEH C18 column (100 mm × 2.1 mm internal dimensions, 1.7 um particle size) on an Acquity UPLC system aligned with a Xevo TQ–S triple stage quadrupole mass spectrometer (Waters) with an ESI source. The mobile phases A and B were water (0.1% formic acid) and acetonitrile/isopropanol (70:30, *v*/*v*), respectively. Concentrations of metabolites were quantitated by Targeted Metabolite identification and quantification (TMBO) software.

### 2.6. Quantitation of BA Metabolites in Serum and Liver

The BAs in rodents’ specimens (serum and liver) were extracted and quantified according to the previous published study [16]. BAs analysis was performed with an ACQUITY BEH C18 column (1.7 mm, 100 mm, 32.1 mm internal dimensions) (Waters, MA, USA) on the instrument SCIEX Triple Quad 5500+ LC–MS/MS (AB Sciex, MA, USA). Standard calibration including seven different concentrations (10, 20, 40, 80, 160, 320, 640 nM) and methanol as the blank. The elution solvents were water with 0.1% formic acid (containing 4 mM ammonium acetate) (A) and acetonitrile with 0.1% formic acid/methanol (3:1) (B). The peak annotation and quantitation were performed using Analyst^®^1.6 Software (AB Sciex, MA, USA). Bile acid standards, including cholic acid (CA), chenodeoxycholic acid (CDCA), glycocholic acid (GCA), glycochenodeoxycholic acid (GCDCA), taurochenodeoxycholic acid (TCDCA), deoxycholic acid (DCA), taurodeoxycholic acid (TDCA), glyodeoxycholic acid (GDCA), ursodeoxycholic acid (UDCA), tauroursodeoxycholic acid (TUDCA), and taurolithocholic acid (TLCA) were purchased from Aladdin. Glyoursodeoxycholic acid (GUDCA), 7–ketolithocholic acid (7–KLCA), α–Muricholic acid (αMCA), β–Muricholic acid (βMCA), tauro–α–muricholic acid (TαMCA), and tauro–β–muricholic acid (TβMCA) were purchased from Sigma.

### 2.7. Hematoxylin–Eosin Staining

The mouse livers were fixed in 4% paraformaldehyde (PFA) for at least 24 h, then embedded in paraffin. The 3 µm liver tissue slices were stained according to manufacturer’ protocol using Hematoxylin–Eosin/HE Staining Kit (Solaribio, Beijing, China).

### 2.8. Immunofluorescence

Liver samples were fixed in 4% PFA overnight. Then, the samples were dehydrated with 20% sucrose for 24 h and embedded with OCT. For immunofluorescence staining, 10 µm cryosections were permeabilized and blocked with 5% goat serum dissolved in PBST (PBS with 0.3% Triton X–100) and stained with the indicated antibodies: α–SMA (CST; 19245s; 1:200 dilution), Col I (CST; 720260S; 1:200 dilution), NTCP (Abcam; ab131084; 1:200 dilution), Ki67 (Abcam; ab16667; 1:200 dilution), and Alexa Fluor 546 phalloidin (Invitrogen; A22283; 1:300 dilution). Primary antibodies were incubated overnight at 4 °C. The next day, the slices were stained with the corresponding secondary antibody conjugated to Alexa Fluor 488 (Beyotime; A0432; 1:250 dilution) or Cy3 (Beyotime; A0516; 1:250 dilution). Nuclei were labeled with DAPI for 5 min (Beyotime; P0131) and rinsed with PBST. The pictures were acquired with a conformal fluorescence microscope (Leica Microsystems, SP–8). 

### 2.9. Flow Cytometry

Liver tissues were harvested and stored in the cold DMEM containing 2% fetal bovine serum (FBS) (CellMax, Beijing, China). Single cells suspension was prepared by mechanical grinding dissociation technique. Fixable Viability stain 780 (BD Biosciences, CA, USA) was used to exclude the dead cells and the indicated antibodies were stained for 30 min at room temperature. Flow cytometry was acquired on BD FACSCanto™ II System (BD Biosciences, CA, USA) and data were analyzed by FlowJo software version 10.8.0. Antibodies involved in flow cytometry were as follows: Fixable Viability stain 780 (BD Biosciences, CA, USA); CD16/CD32 (BD Biosciences, 553141); CD45 (Biolegend, 103108); CD11b (BD Biosciences, 550993); F4/80 (BD Biosciences, 566787); LY6C (Biolegend, 128007); LY6G (BD Biosciences, 560601); LY6C/LY6G (BD Bioscience, 553129); CD4 (BD Biosciences, 566407); CD3 (BD Biosciences, 553061); CD8 (BD Biosciences, 553032).

### 2.10. RNA Sequencing and Data Processing

Total RNA was extracted from liver sample by Trizol^®^ Reagent (Invitrogen, CA, USA) and genomic DNA was removed using DNase I (TaKara, Tokyo, Japan), RNA quality was performed using 2100 Bioanalyser (Agilent) and quantified by the ND–2000 (NanoDrop Technologies), only high-quality RNA (OD260/280 = 1.8~2.2 OD260/230 ≥ 2.0, RIN ≥ 6.5, 28S:18S ≥ 1.0, >1 μg) was used for library preparation. RNA–seq library was prepared from 1 μg of total RNA by TruSeq™ RNA sample preparation Kit from Illumina (San Diego, CA, USA), mRNA was isolated according to polyA selection method by oligo(dT) beads and then fragmented by fragmentation buffer. cDNA was synthesized using a SuperScript double-stranded cDNA synthesis kit (Invitrogen, CA, USA) with random hexamer primers (Illumina). Final libraries were size selected for cDNA target fragments of 300 bp on 2% Low Range Ultra Agarose followed by PCR amplified using Phusion DNA polymerase (NEB) for 15 PCR cycles. After quantified samples were sequenced with the Illumina HiSeq xten/NovaSeq 6000 sequencer (2 × 150 bp read length).

The raw reads were trimmed and quality controlled by SeqPrep (https://github.com/jstjohn/SeqPrep, accessed on 15 November 2021) and Sickle (https://github.com/najoshi/sickle, accessed on 15 November 2021) with default parameters. The clean reads were separately aligned to reference genomes with the orientation mode using the HISAT2 software (http://daehwankimlab.github.io/hisat2/, accessed on 15 November 2021) [17]. The mapped reads of each sample were assembled by StringTie (https://ccb.jhu.edu/software/stringtie/index.shtml?t=example, accessed on 15 November 2021) [18]. The differential expression genes (DEGs) were considered significantly different using adjusted *P* < 0.05 and |log_2_ (fold change)| ≥ 1. The gene ontology (GO) enrichment analysis of DEGs was conducted by Sangerbox (http://vip.sangerbox.com/home.html, accessed on 15 November 2021) using log10 (*p*-Value). 

### 2.11. Primary Hepatic Stellate Cells Isolation and Incubation

Primary hepatic stellate cells (pHSCs) were isolated from 12-week-old mice by enzymatic digestion and Percoll density gradient centrifugation. Briefly, liver was perfused in situ with HBSS and digested with pronase E, DNase I, and type IV collagenase (Worthington, LS004188) dissolved in HBSS. Liver tissues were cut into small pieces and transferred to DMEM containing 10% FBS, digested at 37 °C for 15 min. The cells mixtures were filtered through 70 μm mesh and centrifuged at 50× *g* for 3 min at 4 °C. Then, the pellet was discarded, and the supernatant was collected and centrifugated at 550× *g* for 6 min, discarded the supernatant and added 3 mL DMEM and 2mL of 100% Percoll (Solaribo, Beijing, China) to the pellet and vortexed, next gently overlaid with 3 mL of 25% Percoll followed by 2 mL of PBS. This was centrifugated at 900× *g* for 20 min at room temperature. The needle was inserted into the 25% Percoll level to aspirate the cells. Seeding 1 × 10^4^ cells in the confocal dish (NEST, Nanjing, China), 100 μM TCA was added to primary HSCs medium. After 24 h, the medium was collected for further in-vitro study and the cells were stained with α–SMA and Alexa fluor 546 phalloidin as described above.

### 2.12. Cell Culture and Treatment

The HCT116, MC38 and hepatic stellate cell LX2 were obtained from American Type Culture Collection (ATCC, Manassas, VA, USA). CT26 and LoVo were purchased from the Cell Bank of Type Culture Collection of The Chinese Academy of Sciences (Shanghai, China). MC38 and LX2 were cultured in DMEM (Solaribo, Beijing, China), HCT116 were cultured in RPMI 1640 medium (Solaribo, Beijing, China), LoVo and CT26 were cultured in Ham’s F–12K medium (Solaribo, Beijing, China). All media were supplemented with 10% fetal bovine serum, 1% penicillin/streptomycin (Beyotime, Beijing, China). Moreover, 100 µM TCA was added to LX2 and, after 24 h, the LX2 medium and cells protein were collected. 

### 2.13. Transwell Assays

Transwell assays with a 24-well Boyden chamber (Corning, CA, USA) and an 8-μm pore polycarbonate membrane were used to assess cell migration according to the manufacturer’s protocol. Briefly, each group of cells (HCT116/LoVo: 5 × 10^4^, MC38/CT26: 2 × 10^4^ cells/chamber) were seeded in the upper chambers in 200 µL serum-free medium for 24 h while the bottom chamber contained 600 µL conditioned medium as a chemoattractant. Cells that migrated to the underside of the chamber inserts were fixed in PFA for 30 min and stained with 0.1% crystal violet solution for 30 min. 

### 2.14. RNA Extraction and Quantitative RT–qPCR

Total cellular RNA was isolated from 20mg tissue sample using Trizol regent (Invitrogen). RNA for reverse transcription was applied using HiScript^®^ II Q RT SuperMix for qPCR (+g DNA wiper) to generate cDNA (Vazyme, Nanjing, China). Real-time PCR was performed on StepOnePlus Real-Time PCR System (Applied Biosystems, CA, USA) in combination with Cham Universal SYBR qPCR Master Mix (Vazyme, Nanjing, China) according to the manufacturer’s protocol. Gene expression was calculated by the 2^−ΔΔCT^ method after normalization to *Gapdh*. Sequences of primers are listed below (Table 1).

### 2.15. Western Blot 

Proteins were extracted from LX2 cells and lysed using RIPA lysis buffer with the 10% protease inhibitor PMSF and quantified by BCA protein assay kit (Beyotime, Beijing, China) according to the manufacturer’s instructions. Hence, 40 μg protein were resolved by 10% SDS–PAGE (Hangzi, Shanghai, China) and transferred to PVDF membranes (Millipore, Darmstadt, Germany). Then, membranes were blocked in 5% skim milk in TBST (TBS dissolved with 0.1% Triton X–100) buffer for 1 h at room temperature, followed by incubation with primary antibodies at 4 °C overnight. After incubation with a secondary antibody for 1 h at room temperature, the following antibodies were used in the present study: α–SMA (CST, 19245s, 1:1000 dilution), Col I (CST, 720260S, 1:1000 dilution), FAP (Invitrogen, PA5–99313, 1:1000 dilution), α–tubulin (Proteintech, 11224–1–AP, 1:2000 dilution). Signals were visualized with ECL reagent (Cytiva, MA, USA) and graphed with a Tanon 4600SF Image system (Tanon, Shanghai, China).

### 2.16. Statistical Analysis

Statistical analysis was performed using GraphPad Prism 8.0. Differential metabolites from the serum metabolomic profiling were selected by the False Discovery Rate approach based on the adjusted *P* Value of less than 0.1. Each index comparison between two groups was conducted using unpaired Students’ *t*-test while the one-way ANOVA method was used for multiple group comparison. Comparisons of two or more variables among groups were performed with the two-way ANOVA test. *p* < 0.05 was considered statistically significant differences for comparisons of two or more groups. 

## 3. Results

### 3.1. Altered Bile Acid Profile Is Associated with Liver Metastasis 

Six-week-old C57BL/6J male mice were separately treated with CD and HFD for eight weeks (*n* = 18/groups). Mice were implanted orthotopically with murine cancer cell-derived xenograft (1 mm^3^) at the beginning of the fifth week (Figure 1A). Tumor phenotype showed a higher incidence of liver metastasis in the HFD group (77.8%) relative to the CD group (50%) (Figure 1B). The number of metastatic nodules in the liver of HFD mice was significantly increased (Figure 1C). More tumor foci and lesions were observed in the gross livers and the tissue H&E staining slides from the HFD group (Figure 1D,E). The same diet intervention was performed in a lung-metastatic model with subcutaneous inoculation of 1 × 10^7^ MC38 cells. We found that HFD resulted in a slight increase in the metastatic rate with no change in the number of lung nodules (Appendix A). The data indicate that HFD tends to promote CRLM, with much less effect on lung metastasis. 

Upon HFD feeding, the weights of body, inguinal, and epididymal white adipose tissues were raised in models (Figure 1F–H), accompanied by higher levels of serum total cholesterol (TC), low-density lipoprotein cholesterol (LDL–C), and high-density lipoprotein cholesterol (HDL–C). There was no difference in the weight of brown adipose tissues and serum total triglycerides (TG) level. A targeted metabolomic profiling was further analyzed in the serum of CD or HFD–treated orthotopic models using a liquid chromatography with tandem mass spectrometry (LC–MS/MS) –based method. Results presented an evidently differential serum metabolome pattern between CD and HFD groups, which is largely attributed by a group of bile acids (BAs) that were significantly increased in HFD mice (Figure 1I,J and Appendix A). The above results suggest that liver metastasis is associated with a dysregulated lipid metabolism, particularly an excess of bile acids.

### 3.2. Increased Hepatic TCA Level and Ntcp Expression in Liver-Metastatic Mice

To investigate how BA metabolism correlates with liver metastasis, serum and hepatic BAs were quantified in CD or HFD–treated mice with a strict pathological confirmation of tumor focus (*n* = 9 groups, four non-metastasis and five liver metastasis for each group). The total BA levels in non-metastatic (NM) mice were unchanged between CD and HFD groups, whereas liver metastatic (LM) mice, especially those fed with HFD, had higher total BA levels than NM mice (Appendix A). Such increased total BAs of LM mice were attributed by a group of primary BAs, which were overexposed in the liver as conjugates (CPBAs), while in the serum as the free type (FPBAs) (Figure 2A,B).

Taurocholic acid (TCA) as the most predominant CPBA was raised in the serum and liver of LM mice, showing a higher level in the HFD group relative to the CD group (Figure 2C,E). Other CPBAs, such as tauro–α/β–muricholic acid (Tα/βMCA) and glycocholic acid (GCA), were elevated in the HFD–LM group only. Of FPBAs, cholic acid (CA) and chenodeoxycholic acid (CDCA) showed higher levels in serum of LM mice, and α– and β–muricholic acid (α– and β–MCA) were aberrantly increased in serum of HFD–LM mice (Figure 2D,F). 

Moreover, HFD resulted in an excess of free and conjugated secondary BAs (FSBAs and CSBAs), e.g., taurodeoxycholic acid (TDCA) and deoxycholic acid (DCA), in the serum and liver of LM mice, with lowering free BA levels in NM mice (Figure 2A–F). The correlation analysis revealed that the two conjugated BAs TCA and TDCA had positive relationships with liver tumor load (Figure 2G). The above data suggest that TCA is associated with liver metastasis whether given CD or HFD, while TDCA is more likely linked with HFD–induced liver metastasis.

BA metabolic genes were determined in the liver and ileum. There was no difference in ileal BA transporters (*Asbt* and *Ostα/β*), but there was a significant elevation in the mRNA expression of BA receptor farnesoid X receptor (Fxr), downstream effector small heterodimer partner (Shp) and fibroblast growth factor 15 (Fgf15) in HFD mice (Appendix A). While HFD resulted in different hepatic gene changes in mice between NM and LM. The mRNA level of the BA synthase cytochrome P450 7A1 (Cyp7a1) was significantly reduced in HFD–NM mice relative to CD–NM mice, whereas other BA synthases cytochrome P450 8B1 (*Cyp8b1*) and cytochrome P450 27A1 (*Cyp27a1*) showed higher gene expressions in HFD–LM mice (Figure 2H). Moreover, BA transporter Na^+^–taurocholate cotransporting polypeptide (*Ntcp*), organic anion transporting polypeptide (*Oatp1*), bile salt export pump (*Bsep*), BA receptor farnesoid X receptor (*Fxr*), and its downstream effector small heterodimer partner (*Shp*) had increased gene expressions in HFD–LM mice. Notably, *Ntcp* was highly expressed in HFD and CD–LM mice, and the HFD–LM group showed a more increase than the CD–LM group (Figure 2H), suggesting a higher capability of hepatic uptake of conjugated BAs for LM mice. 

The above results indicate that the BA metabolic change varied between NM and LM mice with HFD feeding. NM mice exhibited a suppressive hepatic BA synthesis/disposition but an enhanced hepatic uptake of conjugated BAs from circulation; while LM mice had higher levels of BA uptake/disposition and BA/Fxr sensing. More importantly, whether receiving CD or HFD feeding, a higher level of hepatic uptake, marked by elevated *Ntcp* expression and excessive TCA, was shown to be associated with liver metastasis.

### 3.3. TCA Promotes Liver Metastases and Induces a Tumor-Favorable Microenvironment

A prospective study has reported the positive association between a higher CRC risk and elevated plasma levels of conjugated BAs, with a much higher odds ratio value for TCA than others [19]. TCA was significantly associated with the tumor load in the liver, suggesting its possible contribution role in CRLM. To prove this, mouse models with orthotopic or subcutaneous implantation were intraperitoneally injected (IP) with TCA (30 mg/kg) or PBS for four weeks, and regular chow diets were applied throughout the experiment (Figure 3A and Appendix A). There was an obvious enhancement in the tumor metastatic incidence (50% PBS vs. 90% TCA) in the liver metastatic model whereas TCA had much less effect on lung metastases (Figure 3B and Appendix A). Accordingly, TCA–treated mice had more tumor foci and lesions in the liver (Figure 3C,D). Immunofluorescence (IF) data presented more Ki–67 positive cells in the liver of TCA–treated mice (Figure 3E). Additional metabolic data showed TCA IP injection significantly elevated TCA level in mouse liver without affecting other BAs (Appendix A). Data support that the overexposure of TCA in the liver does indeed contribute to CRLM.

In accordance with the “seed and soil” hypothesis, we supposed that the TCA–exposed liver might provide a favorable microenvironment for disseminated CRC cells. The gene ontology (GO) results revealed an apparent difference in the activated fibroblasts and immune cells between mice treated with PBS and TCA (Figure 3F and Appendix A). Specifically, the marker genes of fibroblast activation (*Acta2*, *Fap*) and fibrosis-related genes (*Fn1*, *Col1a1*, *Col3a1*, *MFap*, *Mmps*, *TGFbs*, *Pdfgrb* etc.) were evidently upregulated in the liver of TCA–treated mice (Figure 3G). Fibroblast activation-related inflammatory cytokines and chemokines showed higher gene expressions in the TCA–treated mice as well (Figure 3H). In addition, hepatic cell fraction data showed an increase of myeloid-derived immune cells, including CD11b^+^Gr–1^+^ cells (myeloid-derived suppressor cells, MDSCs), CD11b^+^Ly6G^+^ cells (granulocytes), and CD11b^+^F4/80^+^ cells (macrophages) in the liver of TCA–treated mice (Figure 3I and Appendix A), whereas the percentages of CD3+ and CD4+ T lymphocytes were lower, possibly due to the elevated proportion of immunosuppressive cells in liver. These results indicate that TCA could induce fibroblast activation and an immunosuppressive microenvironment within the liver, which have been reported as crucial components of a metastatic niche for CRC cell colonization and survival [20,21]. 

### 3.4. TCA–Activated HSCs Enhance the Migration Capability of CRC Cells

Activated fibroblasts can transdifferentiate into highly proliferative and motile myofibroblasts, which are critical components of the tumor-seeding microenvironment [22], and hepatic stellate cells (HSCs), as the main precursors of myofibroblast residing in the liver, are the major contributor to the construction of (pre-)metastatic niches [23,24,25]. The RNA–seq data showed higher gene levels of activated fibroblast markers in the liver of TCA–treated mice suggest that HSCs might be the key mediator to TCA–triggered CRLM. It was confirmed with IF results that α–smooth muscle actin (αSMA) and collagen I (Col I) were highly expressed in the liver of TCA–treated mice (Figure 4A,B). 

To support the role of TCA in HSCs activation and CRC cell metastasis, primary HSCs were isolated from mouse liver and treated with TCA or DMSO for seven days. As shown in Figure 4C, an obvious fiber-like structure with elevated expression of filamentous actin (F–actin) and αSMA were shown in TCA–treated cells but not in DMSO–treated cells. Besides, we collected conditioned medium (CM) of primary HSCs (pHSC–CM) and supplemented the medium of murine CRC cell lines MC38 and CT26, respectively. The transwell assay revealed that TCA–treated pHSC–CM can significantly increase the migration capability in both types of murine CRC cells (Figure 4D,E). To further validate whether such action is available to human-originated HSCs, the effect of TCA on the activation of human cell line LX2 and the migration of human CRC cell lines LoVo and HCT116 were also tested. The Western blot (WB) results found that 24 h of TCA treatment significantly elevated the protein expression of αSMA, Col I, and fibroblast activation protein (FAP) in LX2 cells in a dose-dependent manner (Figure 4F and Appendix A). The conditioned medium of TCA–treated LX2 (LX2–CM) was shown to enhance the migration abilities of the two human CRC cell lines (Figure 4G,H). The above results suggest that TCA–treated liver residing stromal cell HSCs contribute to CRC cell metastasis.

### 3.5. Blockade of TCA Uptake Effectively Attenuates Liver Metastasis

The metabolic change between NM and LM mice revealed a close connection between a high NTCP–mediated BA uptake, TCA, and liver metastasis. To understand whether controlling hepatic BA uptake could inhibit TCA–induced liver metastasis, thereby evaluating its potential as an anti–CRLM target, we blockaded NTCP in a TCA–treated mouse model with polyclonal NTCP anti-mouse neutralized antibody (20 μg/mice twice a week) for two weeks (Figure 5A). Results showed that NTCP was highly expressed in the liver of control and TCA–treated mice, but deficiently presented in mice with neutralization of NTCP (Figure 5B,C). Consistent with the change of NTCP, BA conjugates and TCA levels in the liver were significantly elevated in TCA–treated models compared with control. Both were evidently reduced in NTCP–neutralized mice relative to those treated with TCA alone (Figure 5D,E). The tumor phenotype data illustrated a lower rate of liver metastasis and much lower numbers of liver metastatic nodules in NTCP–neutralized mice relative to TCA–treated mice (Figure 5F,G). 

We also noticed an apparent decrease in the protein expressions of αSMA and Col I, as well as percentages of CD11b^+^Gr–1^+^ MDSCs and CD11b^+^ Ly6G^+^ granulocytes in the liver of NTCP–neutralized mice compared with TCA–treated group (Figure 5H–J). In contrast, the percentages of CD3^+^ T and CD8^+^ T cells were increased. These results indicate that the TCA–induced liver microenvironment composed of activated fibroblasts and recruited immunosuppressive cells can be effectively attenuated following the NTCP neutralization. These results suggest that the blockade of hepatic BA uptake could inhibit TCA–triggered liver metastasis and suppress tumor-favorable microenvironment formation.

## 4. Discussion

It is well-known that BAs are synthesized and conjugated with taurine or glycine in the liver, stored in the gallbladder, and ~95% are reabsorbed from the distal ileum into circulation via the portal vein, consequently returning to the liver, which is known as enterohepatic circulation [26]. As another crucial participant, the gut microbiota is primarily responsible for BA biotransformation (e.g., hydrolysis, dehydroxylation, and epimerization) in the intestinal tract and thus derivate secondary BAs, also controls host BA synthesis by affecting intestinal FXR–mediated feedback signaling [27,28]. 

In terms of the process, HFD stimulates BA secretion to facilitate fat digestion [29,30], which is a major reason that ileal Fxr/Shp/Fgf15 signaling was remarkably increased in mice, regardless of tumor metastasis. Unlike previous findings that HFD hinders BA synthesis [31,32], our data revealed that HFD resulted in a distinct pattern in the hepatic BA synthesis between the NM and LM groups. NM mice exhibited a lowered BA synthetic level (a reduced *Cyp7a1* expression with decrease of free BAs in liver and serum), which is in line with the activated intestinal feedback control signaling. 

Reversely, in the same case of feeding HFD, genes of BA synthases (*Cyp8b1* and *Cyp27a1*) were elevated in the liver of LM mice. Hepatic uptake capability is known to be negatively controlled by BA/FXR/SHP axis as an adaptive response to block excessive BA accumulation in hepatocytes [33]. Again, a higher level of hepatic uptake with increased expression of *Ntcp* shown in LM mice, especially upon HFD feeding, seems to be impossibly explained by the changes in hepatic FXR/SHP signaling. This suggests that there might be additional CRC tumor-derived factors affecting hepatic BA synthesis and uptake in addition to the intestinal or hepatic negative feedback control loop.

The transcription of NTCP and cytochromes P450 (CYPs) is reported to be regulated by hepatocyte nuclear factors (HNFs) [34,35,36]. Particularly, HNF1α and HNF4α are highly expressed in CRC tumors, the former is linked with tumor malignancy of CRC cells and the latter is considered as a critical risk factor for CRC [37,38,39]. HNF4α is known to be a direct trans-activator of NTCP and CYPs [34,35,40]. Given preliminary data supporting that the HNF4α gene was highly expressed in the liver of LM mice (data no shown), we suppose that CRC–derived HNFs might transfer into hepatocytes and thus affect the transcription of NTCP and CYPs via certain carriers (e.g., exosomes). 

The change of BAs, including excess of secondary BAs (e.g., DCA) and conjugated BAs (e.g., TCA), is associated with a higher risk of CRC [19,41,42,43]. The role of BAs in CRC initiation and progression has been well summarized with more attention on secondary Bas [44]. BA–driven tumorigenic mechanisms, are involved in intestinal epithelial destruction [45], reactive oxygen species induction [46], genomic destabilization [47], and cancer stem cell induction [43]. Though the impact of BAs in tumor metastasis is less studied, DCA has been reported to enhance tumor invasiveness through increasing tyrosine phosphorylation of β–catenin [48]. Lithocholic acid (LCA) could weaken the expression of human leukocyte antigen (HLA) class I antigens on the surface of colon cancer cells, which benefits tumor cells to escape immune surveillance [49]. LCA was also found to activate Erk1/2 and in turn, suppress STAT3 phosphorylation to induce IL–8 expression that are implicated in cancer cell invasion and angiogenesis [50]. Nevertheless, the precise role of conjugated BAs on CRC metastasis is yet to be understood. Here, we filled the gap and discovered that, whether feeding HFD or not, excessive TCA could promote CRLM.

A promotive action of conjugated BAs on malignancy has been mentioned before in other cancers. TCA induce the invasive growth of esophageal adenocarcinoma cells through activating sphingosine 1–phosphate receptor 2–mediated yes-associated protein (YAP) [51]. TCA and TDCA can activate YAP and thus induce lymph nodes (LNs) metastasis in a melanoma model [52]. Both TCA and TDCA were positively correlated with the tumor load in our study. In addition to the pro-metastatic effect of TCA proven here, we expect, at the least, that TDCA should contribute to HFD–induced metastasis. The gut microbiota-involved, TDCA–driven pro-metastatic action and mechanism deserves to be further clarified.

Conjugated 12αOH BAs, e.g., TDCA and TCA, serving as activators of HSCs have been pointed to mediate the pathogenesis of liver fibrosis [53]. This work disclosed that TDCA activated HSCs in a G protein-coupled bile acid receptor (GPBAR1, or TGR5)–dependent manner. Our RNA–seq results that *Jak2* and *Stat3* highly expressed in the liver of TCA–treated mice suggest the JAK2/STAT3 signaling activation, which is the downstream of TGR5 and mediator to HSC activation and transdifferentiation [54]. It suggests TCA might activate HSCs by activating TGR5 as well.

The conditioned medium from TCA–treated HSCs promoted the migration of CRC cells, supporting that TCA–activated HSCs mediate liver metastasis. HSC is a liver-specific cell source that is activated and thus transdifferentiated into myofibroblasts, with increasing actin stress fibers and production of growth factors (e.g., PDGF, HGF and TGF–β) and constituents of extracellular matrix (e.g., fibronectin, collagen I and MMPs) [55]. These factors have been proven to enhance tumor cell growth, invasion, and migration capabilities [56]. A published work showed that taurochenodeoxycholic acid (TCDCA) can stimulate the release of TGF–β1 from liver-specific stromal cell Kupffer cells [57], suggesting that such cells may also contribute to TCA–triggered liver tumor microenvironment (TME).

Immune cells, representing another critical component of the TME, benefit the metastasis of malignant cancer cells by intercommunicating with cytokines or chemokines [58]. The role of BAs in immune cell-mediated CRC progression remains undetermined, and the effects of BAs on macrophages and T cells have been reported to affect liver cancer. TCA can induce macrophages polarization into M2–like phenotype and facilitated immunosuppressive TME formation that favors to liver tumor growth [59]. Recruitment of neutrophils and inflammatory monocytes also contributed to the CRC metastasis [60,61,62]. However, the effect of BAs in these immune cells is unclear. In the liver of TCA–injected models, proportions of MDSCs, neutrophils, and macrophages were evidently increased, suggesting that TCA could shape an immunosuppressive TME. 

Additionally, the chemical inhibition of apical sodium-dependent transporter (ASBT) resulted in a marked shift in hepatic BA composition in HFD–fed mice since there was no effect of HFD on *Asbt* [63]. The higher levels of circulating conjugated BAs and intestinal FXR/SHP/FGF15 signaling in HFD–treated metastatic mice than those with regular diets suggest the possibility that an enhanced intestinal transport is involved in HFD–associated liver metastasis. Though intestinal BA metabolism was not focused on in the present work, we cannot rule out the possible contribution of intestinal BA metabolism (active transport and microbiota biotransformation) to CRLM.

## 5. Conclusions

Collectively, this study revealed that abnormal hepatic BA metabolism, particularly enhanced uptake and thus an excess of TCA in the liver, induces HSC activation and TME formation, thus promoting CRLM. However, the limitation of BA trafficking back to the liver with the blockade of NTCP effectively abolished the promotion of TCA on CRLM. We point to a new BA–driven mechanism underlying liver metastases, suggesting the modulation of hepatic BA metabolism limiting the uptake of conjugated BAs as a potential therapeutic strategy against CRLM.

## Figures and Tables

**Figure 1 cells-11-03810-f001:**
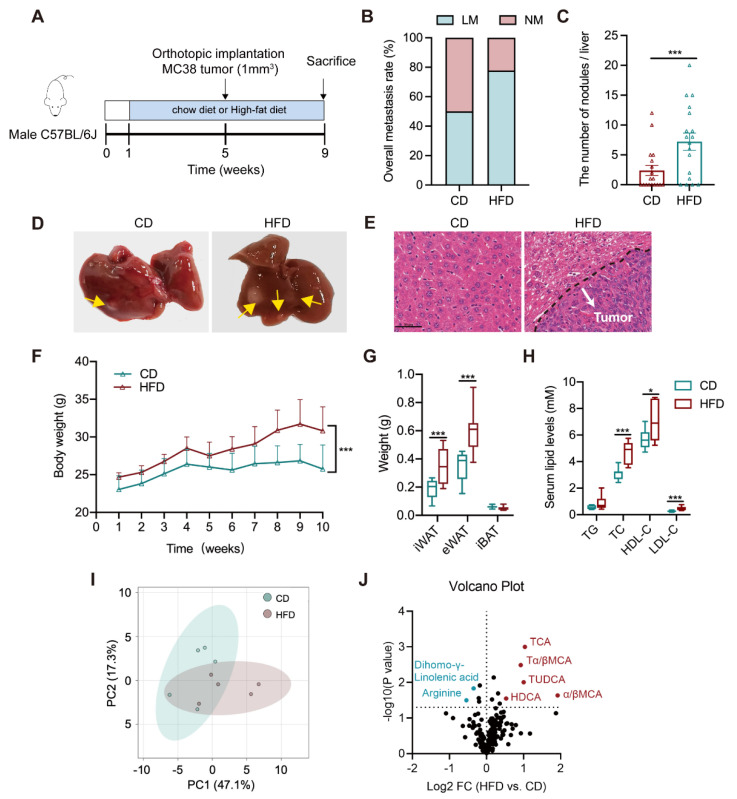
Altered circulating bile acid profile is associated with liver metastasis in an orthotopic implantation mouse model. (**A**) The timeline of the animal experiment (*n* = 18/groups); (**B**) The rate of liver metastasis; (**C**) The number of metastatic nodules in the liver; (**D**) Representative graphs of the gross liver (tumor nodule was labeled with yellow arrow); (**E**) Representative H&E staining images of mouse liver tissue (White arrow: metastatic tumor nodule; Scale bar = 50 μm); (**F**) The weekly body weight during the entire experiment; (**G**) The weights of inguinal white adipose tissue (iWAT), epididymal white adipose tissue (eWAT) and interscapular brown adipose tissue (iBAT); (**H**) Serum levels of triglycerides (TG), total cholesterol (TC), high-density lipoprotein cholesterol (HDL–C), and low-density lipoprotein cholesterol (LDL–C); (**I**) The PCA plot of serum metabolomes between CD and HFD groups (*n* = 5/groups); (**J**) The volcano plot of differential serum metabolites between both groups; The comparisons of each index between CD and HFD groups were conducted with an unpaired Students’ *t*-test, significance was presented as *, *p* < 0.05; ***, *p* < 0.005. Abbreviation: CD, chow diet; HFD, high-fat diet; NM, non-metastasis; LM, liver metastasis; H&E, hematoxylin and eosin; iWAT, inguinal white adipose tissue; eWAT, epididymal white adipose tissue; iBAT, interscapular brown adipose tissue; TG, triglycerides; TC, total cholesterol; LDL–C, low-density lipoprotein cholesterol; HDL–C, high-density lipoprotein cholesterol; PCA, Principal Component Analysis; TCA, taurocholic acid; TDCA, taurodeoxycholic acid; Tα/βMCA, tauro–α/β–muricholic acid, TUDCA, tauroursodeoxycholic acid; α/βMCA, α/β–muricholic acid; HDCA, hyodeoxycholic acid.

**Figure 2 cells-11-03810-f002:**
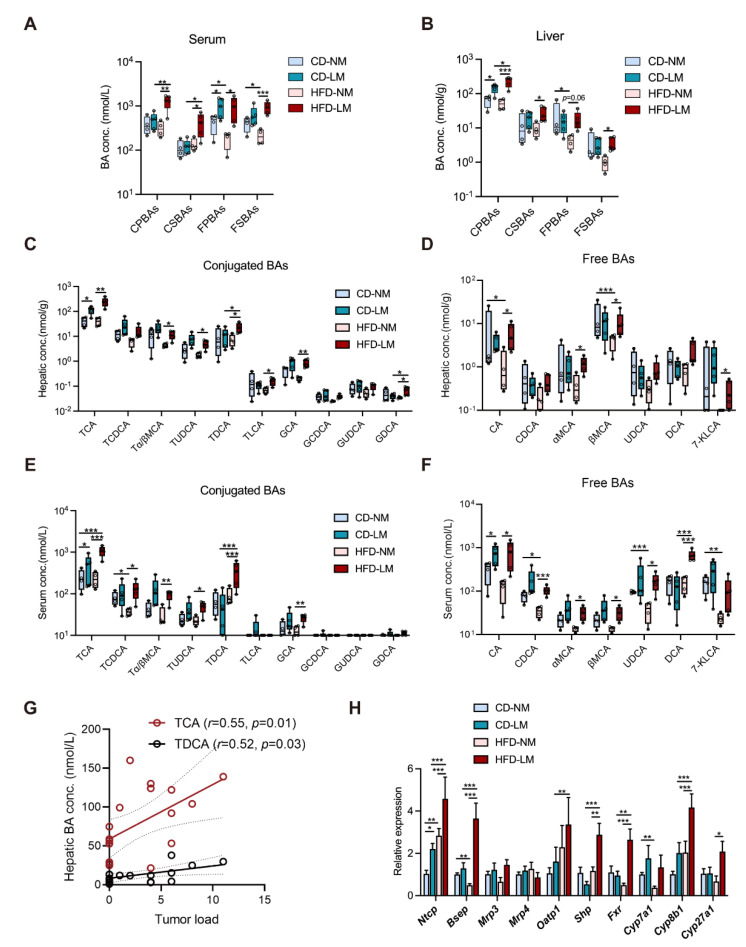
Dysregulated hepatic BA metabolism is strongly associated with liver metastasis in models fed with CD and HFD. (**A**,**B**) The liver and serum BA profiles among LM and NM models from CD and HFD groups (*n* = 9/group, 4 NM and 5 LM); (**C**,**D**) Concentrations of conjugated and free BA species in mouse liver; (**E**,**F**) Concentrations of conjugated and free BA species in mouse serum. (**G**) Pearson’s correlation between concentrations of BAs and tumor load in the liver. (**H**) The relative mRNA expressions of regulators to hepatic BA transport, synthesis, and signaling. Comparisons of BA–related indices among CD or HFD–fed NM and LM groups were determined with a two-way ANOVA test, significance was shown as *, *p* < 0.05; **, *p* < 0.01; ***, *p* < 0.005. Abbreviation: BA, bile acids; CD/HFD–NM, CD or HFD–fed mice without metastasis; CD/HFD–LM, CD or HFD–fed mice with liver metastasis; CPBAs, conjugated primary BAs; CSBAs, conjugated secondary BAs; FPBAs, free primary BAs; FSBAs, free secondary BAs; TCA, taurocholic acid; TCDCA, taurochenodeoxycholic acid; Tα/βMCA, tauro–α/β–muricholic acid; TUDCA, tauro–ursodeoxycholic acid; TDCA, taurodeoxycholic acid; TLCA, taurolithocholic acid; GCA, glycol–cholic acid; GCDCA, glycochenodeoxycholic acid; GUDCA, glycoursodoexycholic acid; GDCA, glycodeoxycholic acid; CA, cholic acid; CDCA, chenodeoxycholic acid; MCA, muricholic acid; UDCA, ursodeoxycholic acid; DCA, deoxycholic acid; 7–KLCA, 7–ketolithocholic acid; *Ntcp*, transporter Na^+^–taurocholate cotransporting polypeptide; *Bsep*, bile salt export pump; *Mrp3*, ATP–binding cassette transporter 3; *Mrp4*, ATP–binding cassette transporter 4; *Oatp1*, organic anion transporting polypeptide; *Shp*, small heterodimer partner; *Fxr*, farnesoid X receptor; *Cyp7a1*, cytochrome P450 7A1; *Cyp8b1*, cytochrome P450 8B1; *Cyp27a1*, Cytochrome P450 Family 27 Subfamily A Member 1.

**Figure 3 cells-11-03810-f003:**
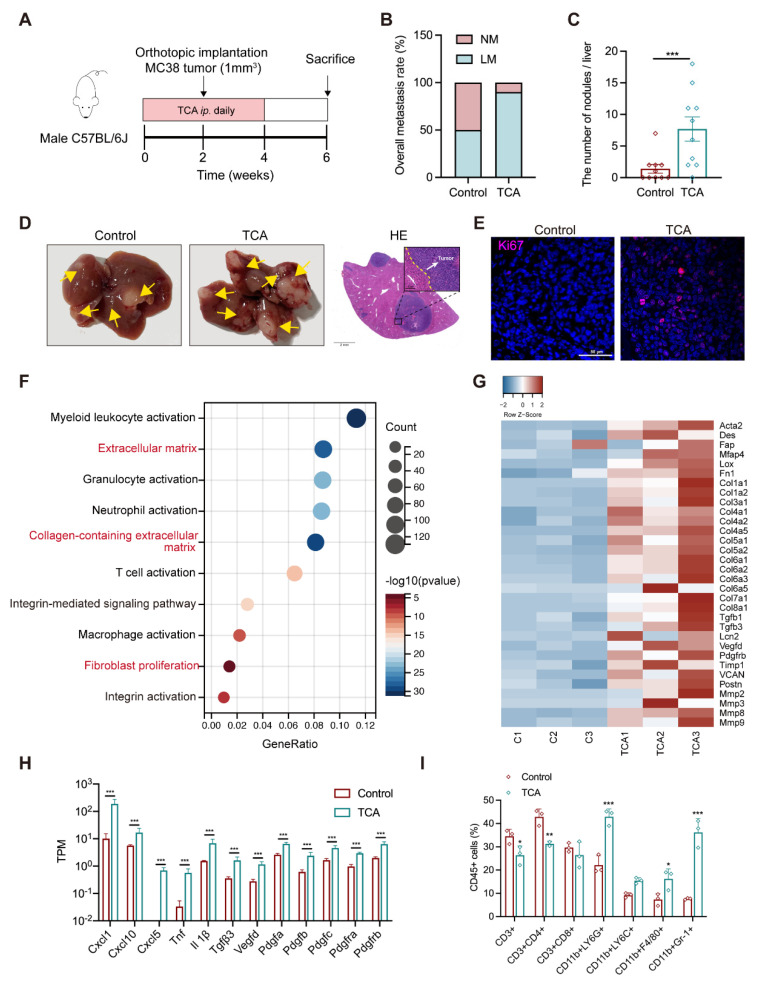
TCA promotes liver metastasis and induces tumor metastatic niche formation in the orthotopic implantation model. (**A**) Timeline of the animal experimental design (*n* = 10/groups); (**B**) The rate of liver metastasis; (**C**) The number of metastatic nodules in the liver; (**D**) Representative gross graphs and H&E staining of the liver (White arrow: metastatic tumor nodule; Scale bar = 2 mm (bottom) or 100 μm (top)); (**E**) Immunofluorescence (IF) staining of Ki67 in mouse liver sections (Purple represents Ki67, Bule represents DAPI; Scale bar = 50 μm). (**F**) The bubble plot of GO analysis of hepatic RNA–seq data between models treated with PBS and TCA (*n* = 3/groups); (**G**) The heatmap of differentially expressed genes related to the activation of fibroblasts between the control and TCA groups (red represents upregulation and blue means downregulation); (**H**) Transcripts per million (TPM) values of myofibroblasts-related chemokines and cytokines; (**I**) Flow cytometry-based counting of hepatic CD45^+^ immune cells, including CD11b^+^ Gr–1^+^ myeloid-derived suppressor cells (MDSCs), CD11b^+^ Ly6G^+^ granulocytes, CD11b^+^ Ly6C^+^ monocytes, and CD11b^+^ F4/80^+^ macrophages, CD3^+^, CD4^+^ and CD8^+^ T cells, (*n* = 3/groups). The statistic of liver metastatic nodules was analyzed with an unpaired students’ *t*-test, RNA–seq data comparison between the control and TCA groups was used with the FDR approach, the adjusted *P* Value of less than 0.5 was considered to differ significantly. Count data from IF or flow cytometry between both groups using unpaired students’ *t*-test, significance was presented by *, *p* < 0.05; ** *p* < 0.01; ***, *p* < 0.005. and significance was expressed by ***, *p* < 0.005. Abbreviation: IP, intraperitoneal injection; NM, non-metastasis; LM, liver metastasis; GO, gene ontology; TPM, transcripts per million.

**Figure 4 cells-11-03810-f004:**
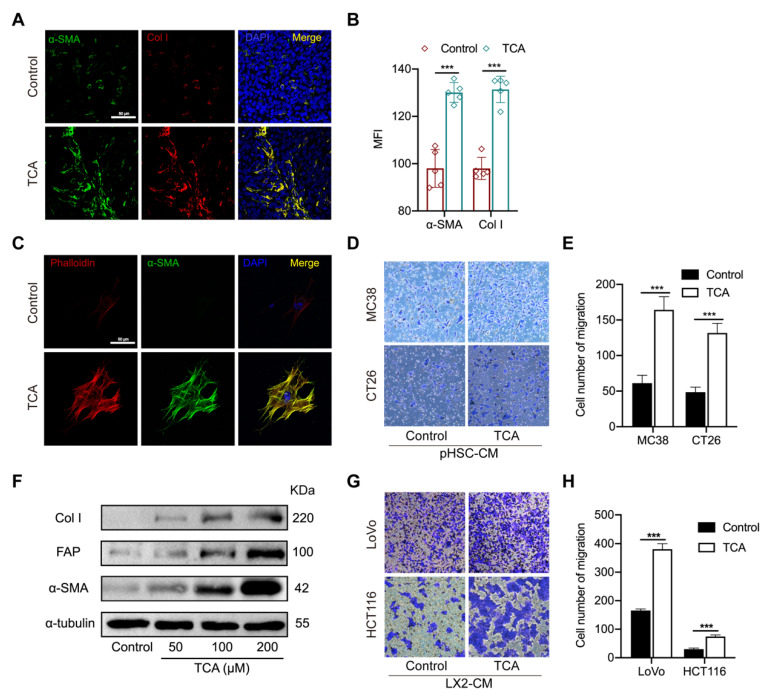
TCA–activated hepatic stellate cells enhance the migrative ability of CRC cells. (**A**,**B**) The IF staining of alpha–smooth muscle actin (αSMA) and collagen I (Col I) in mouse liver sections (Green represent αSMA; Red represents Col I, Bule represents DAPI; Scale bar = 50 μm), the data was calculated by the mean fluorescence intensity (MFI) (*n* = 5/groups); (**C**) The IF staining of αSMA and phalloidin in primary HSCs (pHSCs) treated with TCA (Green represents αSMA; Red represents phalloidin, Bule represents DAPI; Scale bar = 50 μm); (**D**,**E**) The effects of conditioned medium from DMSO or TCA–treated pHSCs on the migration of murine cell lines MC38 and CT26 resulted from the transwell assay; (**F**) Protein expressions of activated HSCs markers αSMA, Col I, and fibroblast activation protein (FAP) in LX2 cells treated with 50, 100, and 200 μM of TCA; (**G**,**H**) The effects of conditioned medium of DMSO or TCA–treated LX2 cells on the migration of human cell lines HCT116 and LoVo resulted from the transwell assay. Comparison between the control and TCA groups was determined using unpaired students’ *t*-test, significance was expressed by ***, *p* < 0.005. Abbreviation: αSMA, alpha–smooth muscle actin; Col I, collagen I; MFI, mean fluorescence intensity; pHSC–CM, the conditioned medium of primary HSCs; αSMA, alpha–smooth muscle actin; FAP, fibroblast activation protein; LX2–CM, the conditioned medium of LX2 cells.

**Figure 5 cells-11-03810-f005:**
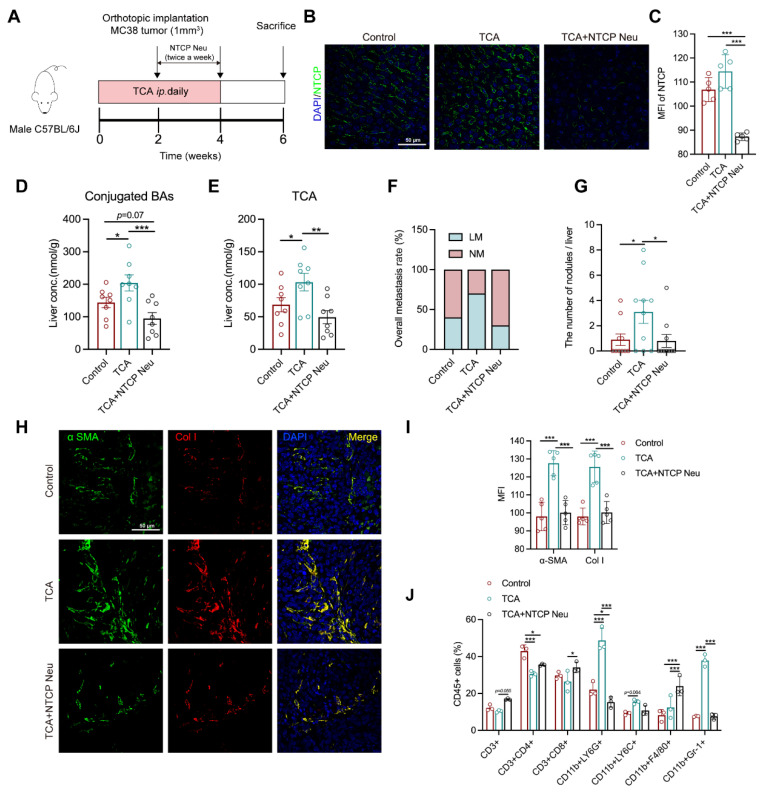
NTCP neutralization can inhibit TCA–induced liver metastasis and the tumor-favorable microenvironment. (**A**) Timeline of the animal experiment (*n* = 10/groups), Na^+^–taurocholate cotransporting polypeptide neutralizing antibody (NTCP Neu) was *IP* injected to models (20 μg/mice) twice a week for 2 weeks; (**B**,**C**) The IF staining of NTCP in mouse liver sections (Green represents NTCP, Blue represents DAPI; Scale bar = 50 μm) and the MFI values of NTCP among groups (*n* = 5/groups). (**D**,**E**) The levels of conjugated BAs and TCA in mouse liver (*n* = 8/groups); (**F**) The rate of liver metastasis; (**G**) The number of liver metastasis nodules; (**H**,**I**) The IF staining of αSMA and Col I in the mouse liver sections (*n* = 5/groups) (Green represents αSMA; Red represents Col I; Bule represents DAPI; Scale bar = 50 μm), the data was calculated by the MFI; (**J**) Flow cytometric analysis of hepatic CD45^+^ immune cells, including CD11b^+^ Gr–1^+^ myeloid-derived suppressor cells (MDSCs), CD11b^+^ Ly6G^+^granulocytes, CD11b^+^ Ly6C^+^ monocytes, CD11b^+^ F4/80^+^ macrophage, CD3^+^, CD4^+^ and CD8^+^ T cells (*n* = 3/groups); The phenotype of the tumors, BAs, fibroblasts, and immune cells were compared among groups using the one-way ANOVA method, significant difference was shown as *, *p* < 0.05; **, *p* < 0.01; ***, *p* < 0.005. Abbreviation: IP, intraperitoneal injection; NTCP Neu, Na^+^–taurocholate cotransporting polypeptide neutralizing antibody; NM, non-metastasis; LM, liver metastasis; αSMA, alpha–smooth muscle actin; Col I, collagen I; MFI, mean fluorescence intensity.

**Table 1 cells-11-03810-t001:** Sequences of paired primers used in the study.

Primers	Sequence (5′−3′)
*Fxr* forward	TGGGCTCCGAATCCTCTTAGA
*Fxr* reverse	TGGTCCTCAAATAAGATCCTTGG
*Shp* forward	CACCTGCATCTCACAGCCACT
*Shp* reverse	GCCAACCCAAGCAGGAAGA
*Ntcp* forward	ATGACCACCTGCTCCAGCTT
*Ntcp* reverse	GCCTTTGTAGGGCACCTTGT
*Bsep* forward	CTGCCAAGGATGCTAATGCA
*Bsep* reverse	CGATGGCTACCCTTTGCTTCT
*Mrp3* forward	CTGGGTCCCCTGCATCTAC
*Mrp3* reverse	GCCGTCTTGAGCCTGGATAAC
*Mrp4* forward	CATCGCGGTAACCGTCCTC
*Mrp4* reverse	CCGCAGTTTTACTCCGCAG
*Oatp1* forward	GTGCATACCTAGCCAAATCACT
*Oatp1* reverse	CCAGGCCCATAACCACACATC
*Cyp7a1* forward	AACAACCTGCCAGTACTAGATAGC
*Cyp7a1* reverse	TGTAGAGTGAAGTCCTCCTTAGC
*Cyp8b1* forward	GGCTGGCTTCCTGAGCTTATT
*Cyp8b1* reverse	ACTTCCTGAACAGCTCATCGG
*Cyp27a1* forward	GCCTCACCTATGGGATCTTCA
*Cyp27a1* reverse	TCAAAGCCTGACGCAGATG
*Asbt* forward	GGAACTGGCTCCAATATCCTG
*Asbt* reverse	GTTCCCGAGTCAACCCACAT
*Ostα* forward	GCCAGGCAGGACTCATATCAAA
*Ostα* reverse	GGCAACTGAGCCAGTGGTAAGA
*Ostβ* forward	CAGGAACTGCTGGAAGAAATGC
*Ostβ* reverse	GCAGGTCTTCTGGTGTTTCTTTGT
*Gapdh* forward	CGACTTCAACAGCAACTCCCACTCTTCC
*Gapdh* reverse	TGGGTGGTCCAGGGTTTCTTACTCCTT

## Data Availability

The data that support the findings of this study are available from the corresponding author upon reasonable request. All sequence data sets are available in the NCBI BioProject database under accession number PRJNA901071.

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
