# Peer review of "A High Hepatic Uptake of Conjugated Bile Acids Promotes Colorectal Cancer—Associated Liver Metastasis"

_cells, 2022, doi:10.3390/cells11233810_

Round 1

Reviewer 1 Report

Experiments in general are well designed and results presented are interesting, but some conflicting data are are difficult to explain. The study is descriptive, lacking mechanistic insight of bile acids to liver metastasis. 

Major concerns:

1. TCA fed or ip to mice are deconjugated by gut bacteria, converted to DCA, which is transported to the liver and conjugated  to TDCA. The amount of TCA and CA in mice are higher than DCA and TDCA, but TDCA is much more hydrophobic and  toxic than TCA.  Person's correlation (Fig. 2G) are very similar between TCA and TDCA. The conclusion that TCA triggered liver metastasis is not proven. T

2. HFD is known to increase bile acid synthesis. Total bile acid pool size (bile acids in liver, gallbladder and intestine) should be determined. 

2. In vitro experiments in isolated cells, authors should compare effects of TCA to TDCA. Lower concentrations of bile acids should be used to reduce toxicity. 

3. Bile acids are known to inhibit NTCP expression (via RAR) but induce BSEP (a FXR target) expression in enterohepatic circulation of bile acids. Data in FIg. 2H showed parallel increase on expression of these two transporters by HFD. NTCP is coupled to Na+ uptake.  OATPs are induced, may be also responsible for bile acid uptake into liver. 

4. Increased FXR/SHP by HFD should inhibit Cyp8b1, not induce Cyp8b1 mRNA shown in Fig. 2F.  FXR/SHP mechanism may be more important in inhibiting Cyp8b1 than Cyp7a1, while Intestinal FGF15/FGFR5/betaKlotho signaling may be more important in inhibiting CYP7a1, than Cyp8b1. It is important to study intestinal ASBT, SHP, FGF15, Osta/OSTb expression, since the gut to liver axis plays a critical role in liver metabolism. 

5. What is the mechanism of increased NTCP expression in mouse tumor models ? 

6. The gut microbiome is totally ignored in this study. At least in the Discussion, authors should describe the role of gut bacteria in secondary bile acid synthesis and regulation of circulating bile acid composition. 

Reviewer 2 Report

The submitted manuscript written Zheng et al. analyses extensively the connection between bile acid effects/level/metabolism and the formation of liver metastasis in colorectal cancer. The manuscript is overall well and clearly written, I have only some suggestions for language editing.

Major problem is that the novelty of the results is not definitely stated all over the manuscript. I suggest to dedicate a paragraph for summarizing what is already known about a BA-driven mechanism of CRC-associated liver metastasis (based on the review of DOI: 10.12998/wjcc.v6.i13.577 , Role of bile acids in colon carcinogenesis, because using only three reference papers (10, 11, 12) are not enough and do not justify the performed experiments). After that highlight what is missing from the big picture and outline the necessity of testing in vivo and in vitro the taurocholic acid effects. 

Methodological questions:

In line 288: it is written that “HFD groups (n=9/groups, 4 NLM and 5 LM for each group).” Although originally the HFD group contained 14 mice. How was selected 9 mice for hepatic BA profile analysis and for further examination out of 14 mice?

Please correct the FACS marker interpretation in the text: CD45 is a pan-hematopoietic cell marker and not only „myeloid leukocyte cell” marker as it is stated in the text.

Figure modifications:

Figure 1B and Figure 3B: Y axis shows the overall metastasis rate not only the liver metastasis %.

Figure 1C, Figure 3C and Figure 5F: please, correct the Y axis as it shows “the number of nodules/ liver”.

Figure 3I and Figure 5J: the CD11 marker is not included in the figures.  Is it correct to use “CD45+ %” on the Y axis? Please check your data accurately.

Please harmonizes the color coding of the columns on two figures: Figure 5J and Figure 3I. The “control” and “TCA” labelling should be the same color on both figures.

Minor mistakes:

Line 56-58 and Line 484-488 contain too long sentences. 

Line 94.: abbreviation should be clarified.

Line 308: remove the dot before the word “only”.

Section 3.3.: please, include that regular chow diet was applied in this experiment.

Line 367: in this sentence “In addition, cell fraction data of flow cytometry found increase..”  please, include for better reading that „hepatic cell fraction data”.
